# Cyclin A/Cdk1 modulates Plk1 activity in prometaphase to regulate kinetochore-microtubule attachment stability

Ana Maria G Dumitru[1,2], Scott F Rusin[1,2], Amber E M Clark[1,2], Arminja N Kettenbach[1,2], Duane A Compton[1,2]*

[1]Department of Biochemistry and Cell Biology, Geisel School of Medicine at Dartmouth, Hanover, United States; [2]Norris Cotton Cancer Center, Lebanon, United States

**Abstract** The fidelity of chromosome segregation in mitosis is safeguarded by the precise regulation of kinetochore microtubule (k-MT) attachment stability. Previously, we demonstrated that Cyclin A/Cdk1 destabilizes k-MT attachments to promote faithful chromosome segregation. Here, we use quantitative phosphoproteomics to identify 156 Cyclin A/Cdk1 substrates in prometaphase. One Cyclin A/Cdk1 substrate is myosin phosphatase targeting subunit 1 (MYPT1), and we show that MYPT1 localization to kinetochores depends on Cyclin A/Cdk1 activity and that MYPT1 destabilizes k-MT attachments by negatively regulating Plk1 at kinetochores. Thus, Cyclin A/Cdk1 phosphorylation primes MYPT1 for Plk1 binding. Interestingly, priming of PBIP1 by Plk1 itself (self-priming) increased in MYPT1-depleted cells showing that MYPT1 provides a molecular link between the processes of Cdk1-dependent priming and self-priming of Plk1 substrates. These data demonstrate cross-regulation between Cyclin A/Cdk1-dependent and Plk1-dependent phosphorylation of substrates during mitosis to ensure efficient correction of k-MT attachment errors necessary for high mitotic fidelity.
DOI: https://doi.org/10.7554/eLife.29303.001

*For correspondence:
duane.a.compton@dartmouth.
edu

**Competing interests:** The authors declare that no competing interests exist.

## Introduction

Mitotic cells rely on the precise regulation of the stability of microtubule attachments to kinetochores (k-MT attachments) on chromosomes to ensure genetic fidelity during cell division (*Bakhoum et al., 2009a*; *Bakhoum et al., 2009b*; *Walczak et al., 2010*). Kinetochore-MT attachments must be sufficiently stable to achieve microtubule occupancy at kinetochores to move chromosomes and to satisfy the spindle assembly checkpoint (SAC) in a timely manner (*Bakhoum et al., 2009b*; *Bakhoum and Compton, 2012*; *Musacchio and Salmon, 2007*; *McEwen and Dong, 2009*). At the same time, microtubules must rapidly detach from kinetochores to promote efficient correction of erroneously oriented k-MT attachments (*Thompson et al., 2010*; *Maiato et al., 2004*). Indeed, hyperstable k-MTs are an underlying cause of the elevated frequency of chromosome mis-segregation associated with chromosomal instability (CIN) observed in a majority of aneuploid tumor cells (*Thompson et al., 2010*; *Cimini and Degrassi, 2005*; *Cimini, 2008*; *Weaver and Cleveland, 2007*; *Bakhoum et al., 2009a*; *Bakhoum et al., 2009b*). Systematically destabilizing k-MT attachments was shown to be sufficient to restore faithful chromosome segregation to tumor cells that otherwise exhibited exceptionally high chromosome mis-segregation rates (*Bakhoum et al., 2009b*; *Kabeche and Compton, 2013*; *Stolz et al., 2010*). Thus, there is a causal relationship between the stability of k-MT attachments and the propensity for chromosome mis-segregation.

Kinetochores rely on several molecular complexes to bind microtubules, and reversible protein phosphorylation is often used as a mechanism to regulate the MT binding affinity of those

complexes to ensure that k-MT attachment stability falls within an acceptable range needed for genomic stability (*Olsen et al., 2010*; *Bakhoum and Compton, 2012*; *Bakhoum et al., 2009b*; *Rogers et al., 2016*; *Kettenbach et al., 2011*)(reviewed in [*Godek et al., 2015*]). For example, Polo-like kinase 1 (Plk1) is an essential mitotic kinase that regulates many aspects of mitosis (extensively reviewed in [*Schmucker and Sumara, 2014*; *Petronczki et al., 2008*; *van Vugt and Medema, 2005*]), including the stabilization of k-MT attachments in prometaphase (*Lénárt et al., 2007*; *Liu et al., 2012*; *Moutinho-Santos et al., 2012*; *Suijkerbuijk et al., 2012*; *Hood et al., 2012*; *Maia et al., 2012*; *Li et al., 2010*). Conversely, Aurora B kinase serves an evolutionarily conserved role for the correction of k-MT attachment errors by destabilizing k-MT attachments (*Welburn et al., 2010*; *Salimian et al., 2011*; *DeLuca et al., 2011*; *Lampson and Cheeseman, 2011*; *Cimini et al., 2006*; *Chan et al., 2012*). In addition, Cdk1 acts in conjunction with either Cyclin A or Cyclin B to drive cell cycle progression into and through mitosis (*Parry et al., 2003*; *Vázquez-Novelle et al., 2014*). Recently, we demonstrated that Cyclin A/Cdk1 plays a role in destabilizing k-MT attachments to promote faithful chromosome segregation (*Kabeche and Compton, 2013*). Similarly, Cyclin A2 has been recently shown to promote microtubule dynamics during meiosis II in mouse oocytes to decrease the frequency of merotelic attachments and the incidence of lagging chromosomes (*Zhang et al., 2017*). These data revealed a catalytic role for Cyclin A/Cdk1 during prometaphase, but the substrates being specifically phosphorylated during prometaphase remain unknown. To address this deficiency, we have devised a strategy using mass spectrometry to identify substrates of Cyclin A/Cdk1 in mitotic human cells arrested in prometaphase.

## Results

### Mitotic Cyclin A/Cdk1 phosphoproteome

To identify Cyclin A/Cdk1-specific mitotic targets in prometaphase, we utilized Stable Isotope Labeling of Amino Acids in Cell Culture (SILAC) (*Ong et al., 2002*) to conduct a bioinformatics-assisted phosphoproteomic screen comparing phosphorylation of protein peptides from cells arrested in mitosis with high versus low levels of Cyclin A. RPE1 cells were metabolically labeled and arrested in prometaphase using the microtubule-stabilizing compound taxol. Mitotically-arrested cells from each population were collected by mechanical 'shake-off.' The heavy-labeled population of cells was immediately flash-frozen (*Figure 1A*). The light-labeled population of cells was incubated for an additional 10 hr in taxol-containing media prior to being flash-frozen (*Figure 1A*). This prolonged mitotic arrest was sufficient for the selective degradation of Cyclin A (*den Elzen and Pines, 2001*; *Geley et al., 2001*; *Di Fiore and Pines, 2010*) without premature mitotic exit as judged by the persistence of substantial levels of Cyclin B and securin (*Figure 1B*) (*Brito and Rieder, 2006*). Furthermore, this strategy did not substantially change the quantities of other known mitotic regulators including Plk1, Nek2, or Aurora B kinase, but did show an increase in phospho-Chk2, an indicator of a DNA damage response pathway (*Figure 1B*) (*Matsuoka et al., 2000*). Thus, these represent mitotic cell populations with high and low Cyclin A levels, respectively (*Figure 1A*).

The differentially labeled high and low Cyclin A-expressing populations of cells were individually lysed and equal quantities of protein were mixed and digested into peptides. Phosphopeptides were enriched using titanium dioxide ($TiO_2$), fractionated and analyzed by LC-MS/MS (*Figure 1A*). In four replicate analyses, we identified 41,892 phosphorylation sites on 6288 proteins of which 30,874 phosphorylation sites on 5346 proteins were quantified (*Supplementary file 1*; *Figure 1C*). 12,609 of these phosphorylation sites contain the minimal Cdk1 consensus motif (pS/pT-P; [*Nigg, 1993*; *Moreno and Nurse, 1990*; *Songyang et al., 1994*]) of which 1264 of these phosphorylation sites with the Cdk1 minimal consensus motif were at least two-fold (average $\log_2$ H/L ratio $\geq$1) more phosphorylated in the high Cyclin A condition compared to the low Cyclin A condition (*Figure 1C*). Of these phosphorylation sites 135 were reproducibly quantified with a p-value of <0.1. Proteins with these phosphorylation sites were additionally filtered for known mitotic roles, which identified 300 (47 with p-value<0.1) unique phosphorylation sites on 156 proteins (30 proteins with p-value<0.1) (*Figure 1C*). 118 (22 p-value<0.1) of these phosphorylation sites on proteins with known mitotic function also contained additional cyclin specificity determinants for an optimal Cdk1 consensus motif (K or R at +2/+3) (*Figure 1C* and *Supplementary file 1*) (*Errico et al., 2010*; *Alexander et al., 2011*).

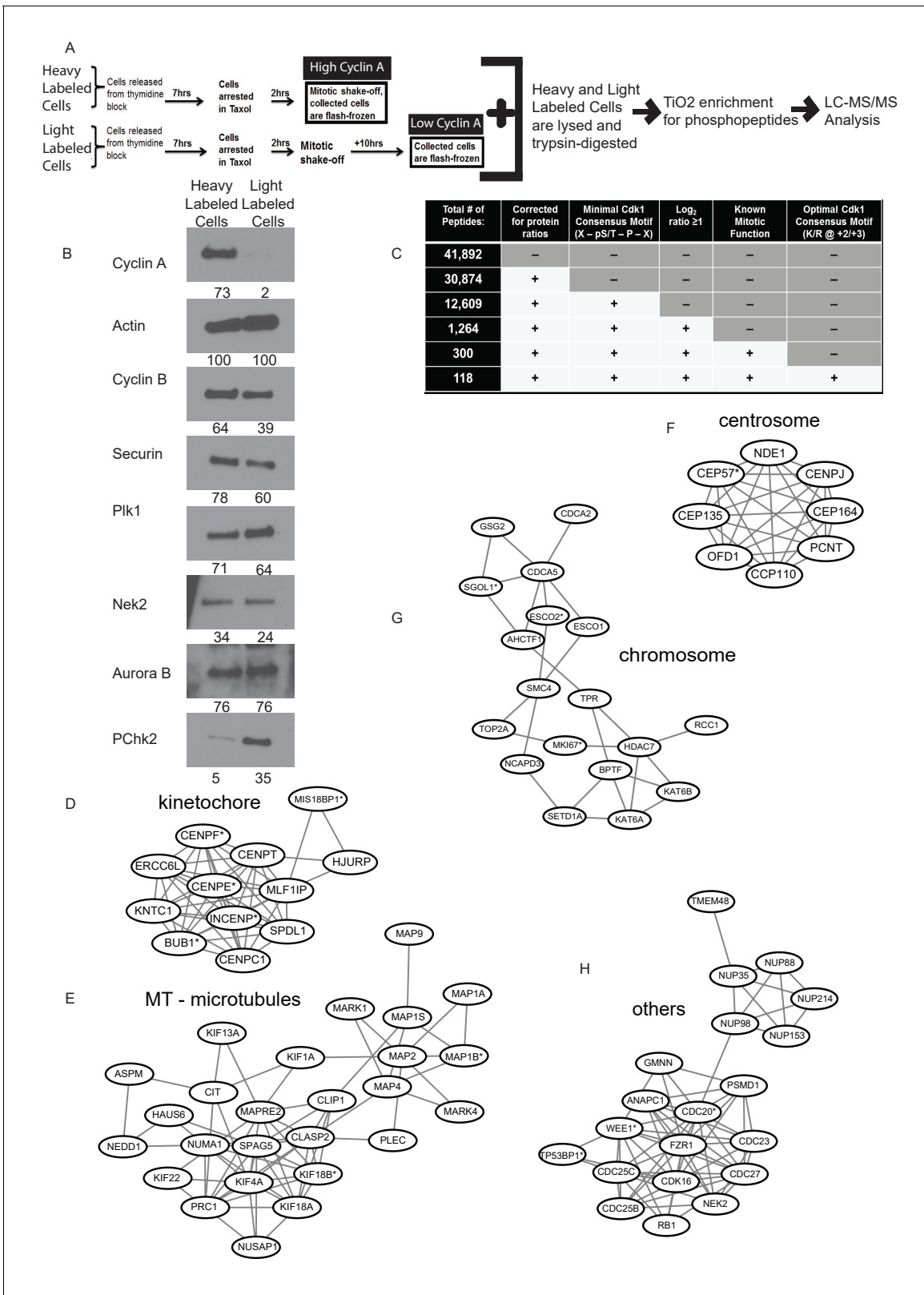

**Figure 1.** Identification of Cyclin A/Cdk1 substrates in prometaphase cells using SILAC-MS/MS. (**A**) Experimental design for SILAC approach to generate two independently labeled prometaphase cell populations with differential Cyclin A levels for analysis of phosphorylation changes. (**B**) Western blots using whole cell lysates from the independently labeled prometaphase cell populations for the indicated protein targets. Numbers indicate protein levels relative to actin loading control in each column. (**C**) Summary of phosphopeptides identified by this proteomic approach. (**D**) –(**H**)
*Figure 1 continued on next page*

*Figure 1 continued*

Protein interactions within different mitotic categories based on STRING database analysis and drawn using Cytoscape. *indicates p<0.1 for at least one phosphopeptide on the corresponding protein.

DOI: https://doi.org/10.7554/eLife.29303.002

The following figure supplements are available for figure 1:

**Figure supplement 1.** Triaged Cyclin A/Cdk1 Prometaphase Substrates by Category.

DOI: https://doi.org/10.7554/eLife.29303.003

**Figure supplement 2.** Triaged Aurora B Prometaphase Substrates by Category.

DOI: https://doi.org/10.7554/eLife.29303.004

**Figure supplement 3.** Triaged Plk1 Prometaphase Substrates by Category.

DOI: https://doi.org/10.7554/eLife.29303.005

---

As expected, not all known Cdk1 phosphorylation sites identified in this phosphoproteome displayed a significant change in phosphorylation under these conditions. For example, histone H3-like centromere protein A (CENP-A) has closely spaced Cdk1 consensus motifs at Ser17 and Ser19 that were unchanged between the high and low Cyclin A conditions (*Kunitoku et al., 2003*). Another example is Lamin B2, a component of the nuclear lamina which has been shown to be phosphorylated by Cyclin B/Cdk1 in vitro and in vivo (*Lüscher et al., 1991*). Lamin B2 has two closely spaced Cdk1 consensus motifs at Thr14 and Ser17 that are sensitive to Cdk1 activity in mitosis (*Petrone et al., 2016*), but they did not significantly change under our conditions.

Furthermore, tyrosine residue 15 on Cdk1, was 6-to-8-fold more phosphorylated in the high Cyclin A condition (*Supplementary file 1*). Phosphorylation of Tyr15 inhibits Cdk1 activity (*Parker and Piwnica-Worms, 1992*), although this negative influence is stronger on Cdk1 (aka cdc2 in those articles) when it is associated with Cyclin B compared to Cyclin A (*Clarke et al., 1992*). Despite any inhibitory influence of Tyr15 phosphorylation, the cells had sufficient Cdk1 activity to have entered and to maintain mitosis under these experimental conditions (*Figure 1B*) (*Swaffer et al., 2016*).

To further elucidate specific Cyclin A/Cdk1 substrates, we performed a bioinformatics approach using the STRING database to interrogate the sorted mitotic proteins that were more than two-fold more phosphorylated at a Cdk1 consensus motif in the high Cyclin A condition for known protein-protein interactions. This revealed several clusters of proteins with previously documented interactions (*Figure 1D–H* and *Figure 1—figure supplement 1*) suggesting that Cyclin A/Cdk1 selectively regulates the proteins in these groups to regulate mitotic functions such as chromatid cohesion and mitotic spindle organization.

## Intersection of mitotic signaling networks

This phosphoproteome showed evidence for cross-regulation between Cyclin A/Cdk1 and other regulators of mitosis including Aurora B kinase, and Plk1, and Protein Phosphatase 1 (PP1). With regard to cross-regulation of Aurora B kinase, we identified several Cyclin A/Cdk1 substrates with established roles in the targeting and/or activation of Aurora B kinase at centromeres. For example, the level of phosphorylation of protein kinase GSG2 (Haspin kinase) was five-fold higher in mitotic cells with high Cyclin A at the Cdk1 consensus site Thr112, and more than two-fold higher at Cdk1 consensus sites Thr97 and Ser147 (*Supplementary file 1*) (*Wang et al., 2010*). Similarly, Bub1 kinase has adjacent Cdk1 sites at Ser661 and Ser665 and the phosphorylation of these sites was more than two-fold higher in our high Cyclin A condition (*Supplementary file 1*) (*Johnson et al., 2004*). Finally, the phosphorylation of the Cdk1 site Ser306 on Inner Centromere Protein (INCENP) increased more than four-fold in the high Cyclin A condition (*Supplementary file 1*) (*Honda et al., 2003*; *Bishop and Schumacher, 2002*). The functional significance of these specific Cdk1 phosphorylation events remains to be determined, but a net effect of these events could be to increase Aurora B kinase localization and/or activity in prometaphase because the extent of autophosphorylation at Thr232 on Aurora B kinase was more than four-fold higher in the high Cyclin A condition (*Supplementary file 1*). (*Wang et al., 2010*), (*Johnson et al., 2004*; *Honda et al., 2003*; *Bishop and Schumacher, 2002*). The overall abundance of Aurora B kinase was not affected by changes in levels of Cyclin A during mitosis (*Figure 1A*) indicating that Cyclin A/Cdk1 activity appears to selectively enhance Aurora B kinase activity in mitosis consistent with parallel roles of

Cyclin A/Cdk1 and Aurora B kinase in destabilizing k-MT attachments (*Welburn et al., 2010*; *Salimian et al., 2011*; *DeLuca et al., 2011*; *Lampson and Cheeseman, 2011*; *Chan et al., 2012*; *Cimini et al., 2006*; *Kabeche and Compton, 2013*).

With regard to cross-regulation of Plk1, we note extensive evidence that INCENP, Bub1 and Haspin regulate Plk1 localization and activity in addition to their regulatory role on Aurora B kinase (*Qi et al., 2006*; *Zhou et al., 2014*; *Goto et al., 2006*). Additionally, the Cdk1 consensus motif containing the dipeptide Ser472:473 on Myosin Phosphatase-Targeting Subunit 1 (MYPT1) was approximately three-fold more phosphorylated in the high Cyclin A condition (p=0.01) (*Supplementary file 1*). The Ser473 site has previously been shown to be phosphorylated by cdc2 in vitro and to be necessary as a priming phosphorylation for Plk1 interaction. MYPT1 functions to link the catalytic subunit of PP1 to Plk1 to locally dampen Plk1 kinase activity by dephosphorylating the activating Thr210 (T-loop) on Plk1 (*Totsukawa et al., 1999*; *Yamashiro et al., 2008*).

With regard to cross-regulation of PP1, the Cyclin A/Cdk1 substrates in this phosphoproteome included the PP1-regulatory subunit PP1-3F and the PP1-catalytic subunit 12C. On PP1-3F, the peptide containing Ser14:Ser18 was seventeen-fold more phosphorylated in the high Cyclin A condition, and on PP1-12C peptides containing Ser399:Ser404, Ser407, Ser509, and Ser604 were each more than two-fold phosphorylated in the high Cyclin A condition (*Supplementary file 1*). The physiological relevance of these phosphorylation events on PP1 activity during mitosis remains to be explored.

These data indicate that some Cyclin A/Cdk1 substrates influence the activities of other mitotic regulators during early mitosis. This finding predicts that this prometaphase phoshoproteome should display not only changes in the phosphorylation status of substrates of Cdk1, but also changes in the extent of phosphorylation of substrates of Aurora B kinase and Plk1 as a consequence of changes in the Cdk1-dependent activity of the aforementioned regulators of Aurora B kinase, Plk1, and PP1. Thus, we interrogated the dataset for phosphopeptides containing the minimal consensus motifs for Aurora B kinase (R/K – X – pS/pT) (*Kim et al., 2010*; *Meraldi et al., 2004*) (*Supplementary file 2*) and Plk1 (D/E-X-pS/pT) (*Supplementary file 3*) (*Nakajima et al., 2003*; *Grosstessner-Hain et al., 2011*).

The majority of phosphorylation sites with consensus Aurora kinase motif did not change between the high and low Cyclin A condition. However, we identified 121 (18 p-value<0.1) phosphorylation sites on proteins with known mitotic function that displayed more than two-fold increase in phosphorylation in the high Cyclin A condition (*Supplementary file 2* and *Figure 1—figure supplement 2*). This is consistent with an increase in Aurora B kinase activity as judged by the phosphorylation of the activating T-loop phosphorylation at Thr232.

Similarly, the majority of phosphorylation sites with Plk1 consensus motif did not change under our conditions. However, we identified 94 (20 p-value<0.1) phosphorylation sites with the Plk1 minimal consensus motif that showed greater phosphorylation in the high Cyclin A condition (*Supplementary file 3* and *Figure 1—figure supplement 3*). In addition, dual specificity protein kinase TTK (MPS1) (*Jelluma et al., 2008*) is in this group of Plk1 substrates that displayed more than three-fold higher phosphorylation in the high Cyclin A condition at a peptide sequence spanning Thr33:Ser37, which contains two sequential Plk1 minimal consensus motifs. Taken together, these data demonstrate cross-regulation between the Cyclin A/Cdk1 and the Aurora B kinase and Plk1 signaling networks in early mitosis, and provide evidence for selective phosphorylation of connected networks of protein substrates as a mechanism of regulating early mitotic spindle and chromosome dynamics.

## Influence of MYPT1 on Plk1 signaling at Kinetochores

To examine the biological significance of this extensive phosphoproteomic dataset in more depth, we selected one Cyclin A/Cdk1 substrate for further analysis. For this purpose, we selected MYPT1 because of its established role in regulating Plk1 activity (*Yamashiro et al., 2008*). To demonstrate that the Cdk1 motif on MYPT1 containing Ser473 is selectively phosphorylated by Cyclin A/Cdk1, we used a phosphosite-specific antibody to quantify the levels of pMYPT1 Ser473 on mitotic chromosomes in the presence and absence of Cyclin A (*Figure 2*). The overall abundance of MYPT1 does not change substantially in cells lacking Cyclin A, but the level of pMYPT1 Ser473 decreases dramatically (*Figure 2C*), and there is a significant reduction in pMYPT1 at centromeres of mitotic chromosomes (*Figure 2A,B*). Importantly, there is very little change in the abundance of Cyclin B/Cdk1 in mitotic cells lacking Cyclin A (*Figure 2C*) indicating that Ser473 on MYPT1 is sensitive to the

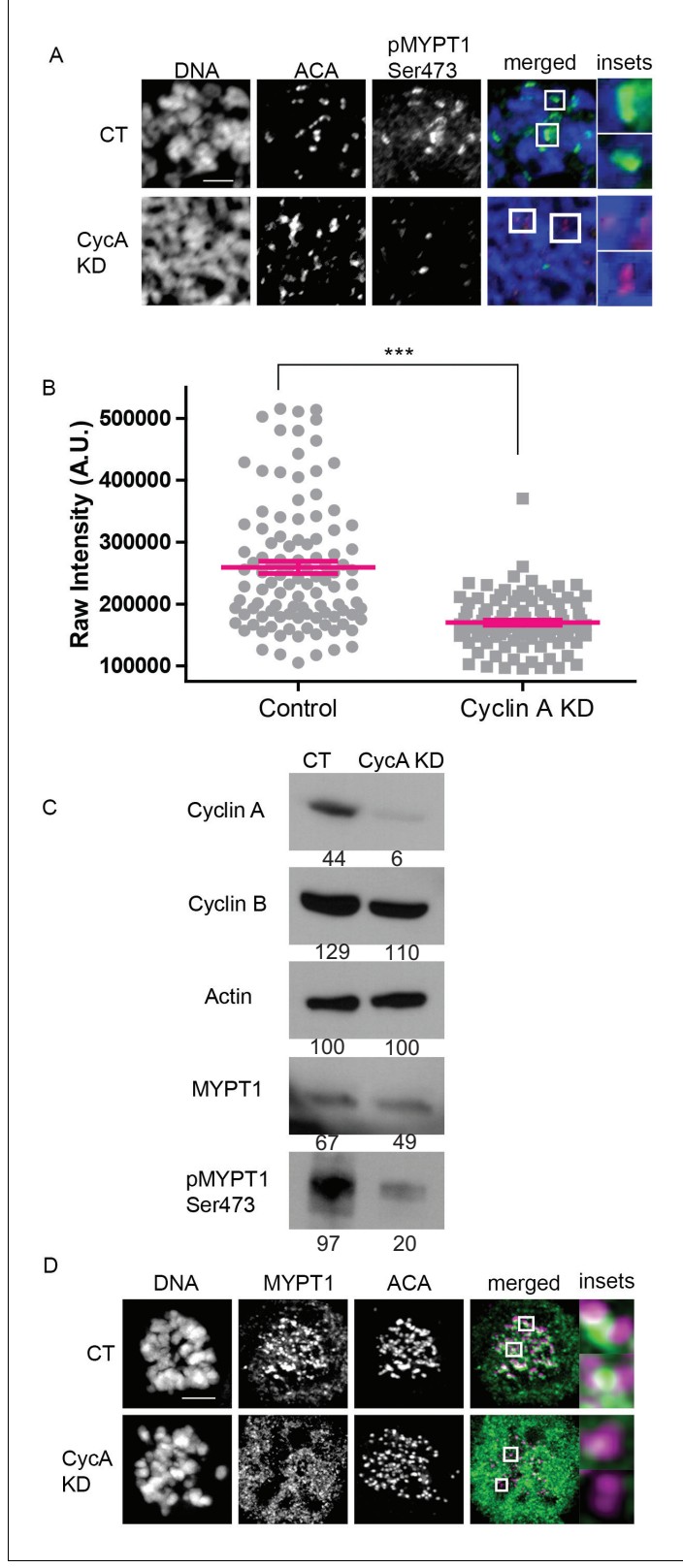

**Figure 2.** MYPT1 is a mitotic Cyclin A/Cdk1 substrate. (**A**) Chromosome spreads from nocodazole-treated U2OS before (CT) and after si-RNA-mediated Cyclin A-knockdown (CycA KD) stained for DNA, centromere-specific human antisera (ACA), antibody specific to MYPT1 pSer473 (pMYPT1 Ser473). Scale bar, 5 μm. Insets highlight centromeres at 3X magnification. (**B**) Quantification of the intensity of centromere MYPT1 pSer473 staining on

*Figure 2 continued on next page*

*Figure 2 continued*

chromosome spreads (n ≥ 100 centromeres/condition. p<0.0001, unpaired, two-tailed t-test). (**C**) Western blots using whole cell lysates from U2OS before (CT) and after si-RNA-mediated Cyclin A-knockdown (CycA KD) for the indicated protein targets. Numbers indicate protein levels relative to actin loading control in each column. (**D**) U2OS before (CT) and after si-RNA-mediated Cyclin A-knockdown (CycA KD) stained for DNA, MYPT1, and centromere-specific human antisera (ACA). Scale bar, 5 μm. Insets highlight centromeres at 5X magnification.

DOI: https://doi.org/10.7554/eLife.29303.006

---

levels of Cyclin A/Cdk1 in mitosis (*Figure 2C*). The localization of MYPT1 to kinetochores was also disrupted by the absence of Cyclin A indicating that Cyclin A/Cdk1 phosphorylation is responsible for targeting MYPT1 to kinetochores (*Figure 2D*). These data demonstrate that Cyclin A/Cdk1 modulates MYPT1 to regulate its localization to kinetochores.

Next, we tested if MYPT1 influenced various aspects of mitosis including the recruitment and/or local activity of Plk1 to kinetochores. With respect to Plk1 localization and activity, the overall abundance is not substantially changed in mitotic cells lacking either Cyclin A (*Figure 1B*) or MYPT1 (*Figure 3—figure supplement 1*). However, quantitative immunofluorescence revealed that the quantity of Plk1 localized to kinetochores increased in cells depleted of MYPT1 (*Figure 3A,B*). This change in Plk1 abundance at kinetochores was detected in prometaphase but not in metaphase (*Figure 3B*). Using a phosphosite-specific Plk1 antibody for Thr210 we detected a substantial increase in phosphorylation on this site in mitotic cells depleted of MYPT1 (*Figure 3—figure supplement 1*). Moreover, we detected an increase in the amount of phosphorylated Plk1 at kinetochores in both prometaphase and metaphase cells lacking MYPT1 (*Figure 3C,D*). These data are consistent with previous reports that MYPT1 recruits the catalytic subunit of PP1 to kinetochores to dephosphorylate the activating loop (Thr210) of Plk1 to dampen Plk1 activity (*Yamashiro et al., 2008*; *Totsukawa et al., 1999*).

Given the established role of Plk1 in regulation of k-MT attachments (*Lénárt et al., 2007*; *Liu et al., 2012*; *Moutinho-Santos et al., 2012*; *Suijkerbuijk et al., 2012*; *Li et al., 2010*; *Hood et al., 2012*; *Maia et al., 2012*), we tested if MYPT1 modulates k-MT attachment stability by influencing Plk1 activity. The stability of k-MT attachments was determined from the quantification of microtubule turnover rates in live cells using fluorescence dissipation after photoactivation (FDAPA) of cells stably expressing photoactivatable GFP-tagged tubulin. The stability of k-MT attachments display a characteristic step-wise increase between prometaphase and metaphase in both RPE1 and U2OS cells (*Figure 4A* and *Figure 4B*, *Figure 4—figure supplement 1*) as we have described previously (*Kabeche and Compton, 2013*). In U2OS cells depleted of MYPT1 we observed an inversion of this step-wise change. In prometaphase U2OS cells lacking MYPT1, k-MT attachments were significantly more stable than in control prometaphase cells (*Figure 4B* and *Figure 4—figure supplement 1*). The stability of k-MT attachments in RPE1 cells was also increased in cells depleted of MYPT1 (*Figure 4A*), although we only report prometaphase since RPE1 cells display chromosome congression defects in the absence of MYPT1 and are delayed in chromosome alignment and anaphase onset.

In contrast to the stabilization of k-MT attachments in U2OS cells lacking MYPT1 in prometaphase, k-MT attachments in metaphase cells lacking MYPT1 were significantly less stable than in control metaphase cells (*Figure 4B* and *Figure 4—figure supplement 1*). Additionally, the k-MT attachments in prometaphase cells lacking MYPT1 were significantly more stable than those of metaphase counterparts. A similar trend was observed by quantifying the intensity of calcium-stable microtubules in mitotic cells (*Figure 4—figure supplement 2*), although it did not reach statistical significance. In addition to these effects, mitotic cells depleted of MYPT1 demonstrated a significant increase in inter-kinetochore distance in both prometaphase and metaphase and a significant increase in spindle length (*Figure 4—figure supplement 3*) indicative of a role of MYPT1 in regulating processes that contribute to the mechanical aspects of mitosis. Importantly, the observed changes in the stability of k-MT attachments were reversed by treatment with low-dose Plk1 inhibitor BI-2536 (*Figure 4B*). This demonstrates that MYPT1 acts to dampen Plk1 activity at kinetochores during prometaphase to ensure the rapid detachment of k-MTs.

To determine the functional significance of phosphorylation at Ser472:473 on MYPT1, we used site-directed mutagenesis to generate mutant versions of MYPT1 where Ser473 was changed to

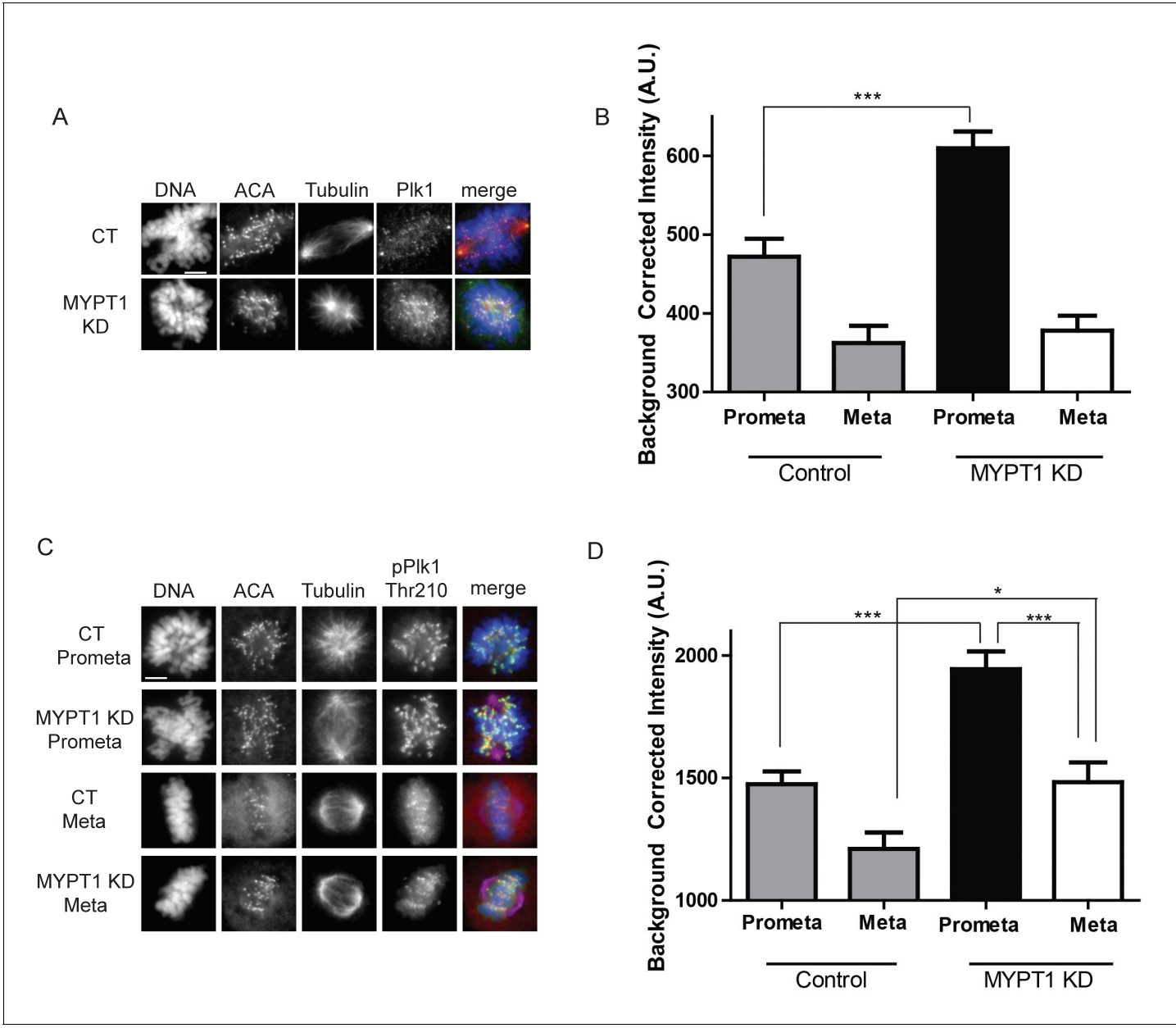

**Figure 3.** MYPT1 limits kinetochore Plk1 localization and phosphorylation. (**A**) U2OS before (CT) and after si-RNA-mediated MYPT1-knockdown (MYPT1 KD) stained for DNA, centromere-specific human antisera (ACA), tubulin, and Plk1 as indicated. Scale bar, 5 µm. (**B**) Quantification of the intensity of kinetochore Plk1 staining in U2OS before (Control) and after si-RNA-mediated MYPT1-knockdown (MYPT1 KD) in prometaphase (Prometa) and metaphase (Meta) cells (n ≥ 100 kinetochores per condition. p<0.0001, unpaired, two-tailed t-test). (**C**) U2OS before (CT) and after si-RNA-mediated MYPT1-knockdown (MYPT1 KD) in either prometaphase (Prometa) or metaphase (Meta) stained for DNA, centromere-specific human antisera (ACA), tubulin, and pPlk1-Thr210 as indicated. Scale bar, 5 µm. (**D**) Quantification of the intensity of kinetochore pPlk1-Thr210 staining in U2OS cells before (Control) and after si-RNA-mediated MYPT1-knockdown (MYPT1 KD) in prometaphase (Prometa) and metaphase (Meta) cells (n ≥ 100 kinetochores per condition. p<0.0001, MYPT1KD prometaphase vs metaphase, p<0.0001, CT vs MYPT1 KD prometaphase; p=0.01 CT vs MYPT1 KD metaphase; unpaired, two-tailed t-test).

DOI: https://doi.org/10.7554/eLife.29303.007

The following figure supplements are available for figure 3:

**Figure supplement 1.** Whole cell lysates compared for changes in protein expression.
DOI: https://doi.org/10.7554/eLife.29303.008
**Figure supplement 2.** Analysis of Plk1 levels.
DOI: https://doi.org/10.7554/eLife.29303.009
**Figure supplement 3.** Analysis of pPlk1 levels.

*Figure 3 continued on next page*

*Figure 3 continued*

DOI: https://doi.org/10.7554/eLife.29303.010

either aspartic acid or alanine and where both Ser472 and Ser473 were changed to aspartic acid. Immunoblot confirms the expression of these mutants and wild type MYPT1 in U2OS cells (*Figure 4—figure supplement 4*). As previously reported, depletion of Cyclin A prematurely stabilized k-MT attachments in prometaphase and eliminated the step-wise increase in k-MT stability between prometaphase and metaphase (*Figure 4C*). Expression of wild type MYPT1 did not alter the step-wise increase in k-MT stability between prometaphase and metaphase, although k-MT attachments were slightly less stable in both phases of mitosis in the presence of excess MYPT1 (*Figure 4C*). Expression of MYPT1-473A reversed the step-wise change in k-MT attachment stability between prometaphase and metaphase (*Figure 4C*) akin to the loss of MYPT1 function (*Figure 4B*). This indicates that alanine mutation at this site acts in a dominant negative fashion to disrupt endogenous MYPT1 function. Expression of either MYPT1-473D or MYPT1-472:473DD abolished the step-wise stabilization of k-MT attachments between prometaphase and metaphase (*Figure 4C*). Cells expressing the aspartic acid mutants of MYPT1 displayed k-MT attachment stability equivalent to control cells, but did not appropriately stabilize k-MT attachments in metaphase. These data suggest that these mutants of MYPT1 are acting appropriately during prometaphase (as a phospho-mimic) and remain persistently active in metaphase to inappropriately destabilize k-MT attachments.

We previously demonstrated a causal relationship between the stability of k-MT attachments and the frequency of chromosome segregation errors (*Kabeche and Compton, 2013*; *Bakhoum et al., 2009b*). Therefore, we tested if the changes in k-MT attachment stability in cells depleted of MYPT1 affected the fidelity of chromosome segregation. The frequency of lagging chromosomes in anaphase was not significantly different between control cells and cells lacking MYPT1 using either U2OS or RPE1 cells (*Figure 4D*). This suggests that in cells lacking MYPT1, the destabilization of k-MT attachments in metaphase can compensate for the hyperstabilization of k-MT attachments in prometaphase to ensure faithful chromosome segregation. However, if we increased the frequency of k-MT attachment errors using the Eg5 inhibitor monastrol, then we observed a significant increase in the frequency of lagging chromosomes in anaphase in cells lacking MYPT1 relative to control cells following the release from the monastrol-induced mitotic arrest (*Figure 4E*). Thus, MYPT1-deficient cells are compromised in correcting k-MT attachment errors under conditions where the frequency of errors is elevated.

The association of Plk1 with substrates is dependent on priming phosphorylation on the substrate, which is frequently carried out by Cdk1 (*Lee et al., 2014*; *Schmucker and Sumara, 2014*; *Kang et al., 2006*; *Kettenbach et al., 2011*; *Elia et al., 2003*). The association of MYPT1 with Plk1 was previously shown to be mediated by priming phosphorylation at Ser473 (*Yamashiro et al., 2008*), and our data indicate that Cyclin A/Cdk1 catalyzes that priming phosphorylation.

Interestingly, we also observed an increase in the quantity of pPlk1 at kinetochores in the absence of MYPT1 (*Figure 3*) and a similar increase in Plk1 and pPlk1 localization to kinetochores in the absence of Cyclin A (*Figure 3—figure supplements 2* and *3*). While Plk1 substrate recognition depends on priming phosphorylation commonly by Cdk1, Plk1 localization to kinetochores is regulated by self-priming phosphorylation of PBIP1 (CENP-U) by Plk1 itself (*Kang et al., 2006*; *Lee et al., 2014*). Indeed, inhibition of Plk1 activity results in its delocalization from kinetochores (*Kang et al., 2011*). Therefore, we examined the phosphorylation of Thr78 on PBIP1. We used a phosphosite specific antibody to measure the levels of pPBIP1 Thr78 in the presence and absence of Cyclin A and MYPT1 (*Figure 5*). Cells depleted of either Cyclin A or MYPT1 displayed an approximate two-fold increase in the level of phosphorylation at Thr78 compared to control cells as judged by immunoblotting of mitotic extracts (*Figure 5A,B*). There was also a significant increase in the localization of pPBIP1 Thr78 to kinetochores in cells lacking either Cyclin A or MYPT1 (*Figure 5C,D*). These data indicate that Cyclin A/Cdk1 promotes MYPT1 targeting to kinetochores where it reduces the level of Plk1 phosphorylation at Thr210 and reduces the level of Plk1 self-priming of PBIP1.

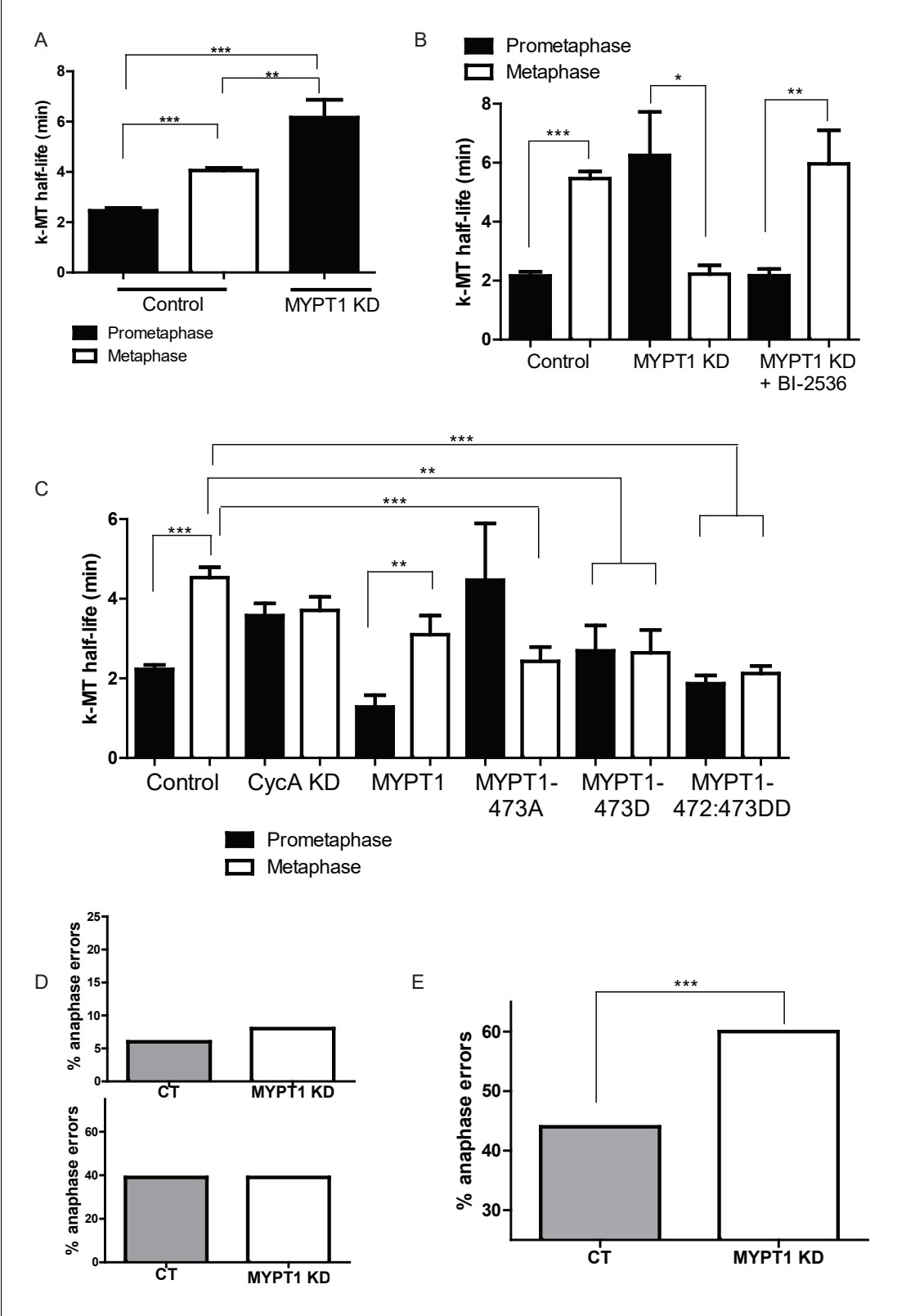

**Figure 4.** MYPT1 promotes efficient error correction by regulating Plk1. (**A**) Microtubule turnover rates were measured and k-MT half-life was calculated from RPE1 cells before (Control) and after si-RNA-mediated MYPT1-knockdown (MYPT1 KD) in prometaphase and metaphase cells as indicated (n ≥ 10 cells/condition from ≥2 independent experiments. ***Indicates p<0.0001, **indicates p=0.0096, unpaired, two-tailed t-test. (**B**) Microtubule turnover rates were measured and k-MT half-life was calculated from U2OS cells before (Control) and after si-RNA-mediated MYPT1-knockdown (MYPT1 KD) and

*Figure 4 continued on next page*

*Figure 4 continued*

with the addition of the Plk1-specific inhibitor Bi-2536 in prometaphase and metaphase cells as indicated (n ≥ 15 cells/condition from ≥2 independent experiments. CT cells p<0.0001; MYPT1 KD cells p=0.04; MYPT1KD +BI-2536 p=0.0038; unpaired, two-tailed t-test). (**C**) Microtubule turnover rates were measured and k-MT half-life was calculated from U2OS cells before (Control) and after si-RNA-mediated Cyclin A2-knockdown (CycAKD) or transfection with plasmid containing either full-length MYPT1 (MYPT1), phosphonull (MYPT1-473A), phosphomimetic (MYPT1-473D) or tandem phosphomimetic (MYPT1-472:473DD) in prometaphase and metaphase cells as indicated (n ≥ 10 cells/condition in all conditions except MYPT1-473A metaphase where n = 9 cells). ***Indicates p<0.0001, **indicates p=0.004, unpaired, two-tailed t-test. (**D**) Fraction of anaphase cells displaying lagging chromosomes in RPE1 (Top) and USOS (Bottom) cells before (CT) and after si-RNA-mediated MYPT1-knockdown (MYPT1 KD) (n ≥ 300 anaphases/condition. p-values n. s., unpaired two-tailed t-test. (**E**) Fraction of anaphase cells displaying lagging chromosomes following release from monastrol treatment in U2OS cells before (Control) and after si-RNA-mediated MYPT1-knockdown (MYPT1 KD) (n ≥ 800 anaphases/condition from two independent experiments. p<0.0001, Chi square contingency analysis).

DOI: https://doi.org/10.7554/eLife.29303.011

The following figure supplements are available for figure 4:

**Figure supplement 1.** Fluorescence Dissipation After Photoactivation (FDAPA) Representative Curves From Microtubule Stability Measurements.
DOI: https://doi.org/10.7554/eLife.29303.012

**Figure supplement 2.** Calcium Stabilization Assay.
DOI: https://doi.org/10.7554/eLife.29303.013

**Figure supplement 3.** Quantifications of Other Mechanical Effects of MYPT1 in Mitotic Cells.
DOI: https://doi.org/10.7554/eLife.29303.014

**Figure supplement 4.** Analysis of Whole Cell Levels of Expression of Various MYPT1 Plasmids.
DOI: https://doi.org/10.7554/eLife.29303.015

## Discussion

The capacity of kinetochores to release their attached microtubules is essential for faithful chromosome segregation (*Nicklas and Ward, 1994*), and we and others have previously established a mitotic role for Cyclin A/Cdk1 in destabilizing k-MT attachments to promote efficient correction of k-MT attachment errors (*Kabeche and Compton, 2013*; *Zhang et al., 2017*). Here, we use a phosphoproteomic approach to identify a large number of substrates of Cyclin A/Cdk1. We provide biochemical validation that one of these substrates, MYPT1, actively participates in destabilizing k-MT attachments in prometaphase by modulating the activity of Plk1 at kinetochores.

These data provide evidence in mammalian cells that the identity of the specific activating cyclin subunit can confer selectivity onto Cdk1 for some substrates (*Pagliuca et al., 2011*; *Loog and Morgan, 2005*; *Errico et al., 2010*; *Moore et al., 2003*; *Brown et al., 1999*; *Bhaduri and Pryciak, 2011*). Whereas many mitotic substrates are likely to be phosphorylated with equivalent efficiency by Cyclin A/Cdk1 or Cyclin B/Cdk1 as indicated by enzyme-substrate interaction data and cyclin-substitution rescue experiments in yeast and HeLa cells (*Pagliuca et al., 2011*; *Swaffer et al., 2016*; *Fisher and Nurse, 1996*; *Gong and Ferrell, 2010*; *Gavet and Pines, 2010*), we show that the phosphorylation of a subset of Cdk1 substrates in mammalian cells relies on the continued presence of Cyclin A in prometaphase. Ongoing phosphatase activity diminishes the level of phosphorylation of these substrates in Cyclin A-deficient mitotic cells, despite the continued presence of Cyclin B/Cdk1 (*Figures 1B* and *2C*). By extension, we would predict that a subset of Cdk1 substrates would also be sensitive to the continued presence of Cyclin B although our phosphoproteomics experiment was not designed to identify that class of substrates should it exist. Thus, we envisage three subsets of mitotic Cdk1 substrates in mammalian cells that are defined by whether they are selectively phosphorylated by Cyclin A/Cdk1, Cyclin B/Cdk1, or non-selectively phosphorylated by either Cyclin A/Cdk1 or Cyclin B/Cdk1; these substrates would display temporal differences in phosphorylation because Cyclin A and Cyclin B are degraded asynchronously during mitosis in mammalian cells (*den Elzen and Pines, 2001*; *Geley et al., 2001*; *Di Fiore and Pines, 2010*; *Brito and Rieder, 2006*).

These data also demonstrate that Cyclin A/Cdk1 catalyzes substrate priming necessary for Plk1 binding (*Figure 6*). Previously, it was assumed that Cdk1-dependent priming of Plk1 substrates in mitosis was provided irrespective of the cyclin binding partner of Cdk1 (*Lee et al., 2014*; *Schmucker and Sumara, 2014*). Our finding that the priming of MYPT1 for Plk1 interaction is sensitive to Cyclin A/Cdk1 reveals a new, and unexpected level of temporal control in the process of Plk1 substrate priming. Cyclin A/Cdk1-catalyzed priming of MYPT1 provides an inherent bias favoring the

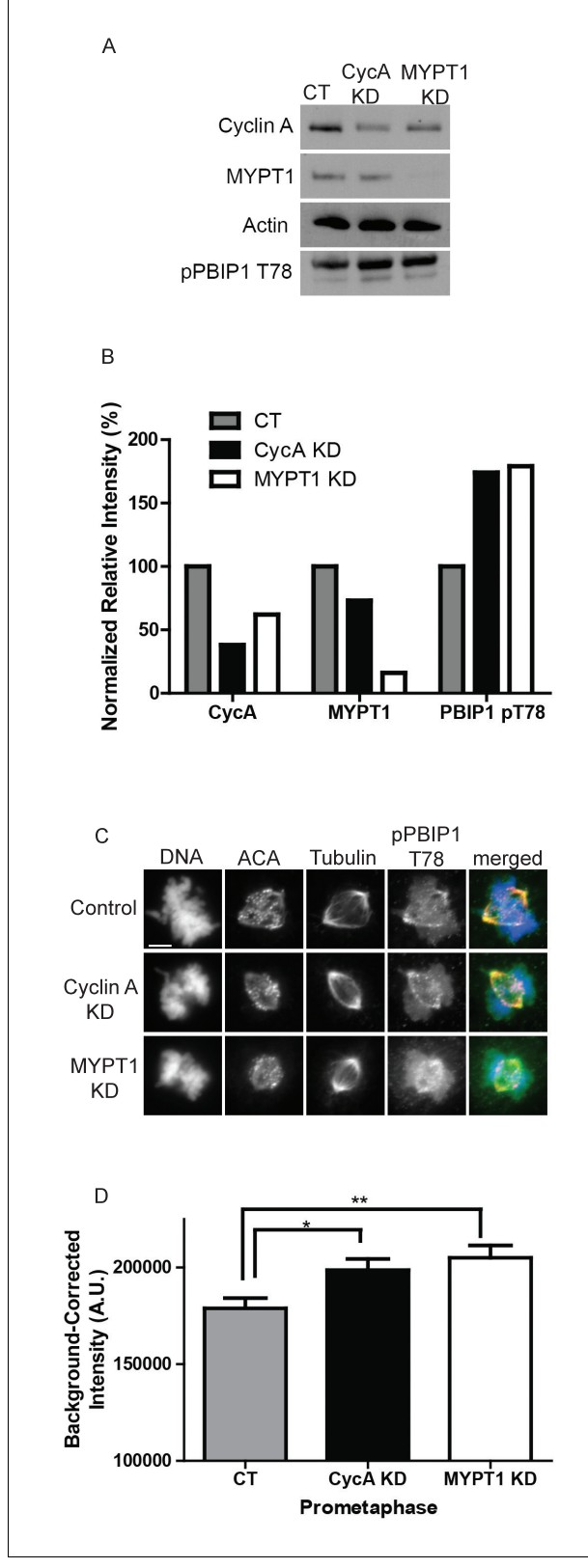

**Figure 5.** Plk1 self-priming of PBIP1 is influenced by Cyclin A/Cdk1 and MYPT1. (**A**) Western blots using whole cell lysates from U2OS before (CT) and after si-RNA-mediated Cyclin A-knockdown (CycA KD) of MYPT1 knockdown (MYPT1 KD) for the indicated protein targets. (**B**) Quantification of Western blots for knockdown efficiency and pPBIP1-Thr78 in the presence or absence of Cyclin A or MYPT1 (p<0.0001, Chi square contingency analysis). (**C**)
*Figure 5 continued on next page*

*Figure 5 continued*

U2OS before (Control) and after si-RNA-mediated Cyclin A knockdown (Cyclin A KD) or MYPT1-knockdown (MYPT1 KD) in prometaphase cells stained for DNA, centromere-specific human antisera (ACA), tubulin, and pPBIP1-Thr78 as indicated. Scale bar, 5 µm. (D) Quantification of the intensity of kinetochore pPBIP1-Thr78 staining in U2OS cells before (CT) and after si-RNA-mediated Cyclin A knockdown (CycA KD) or MYPT1-knockdown (MYPT1 KD) in prometaphase cells (n ≥ 100 kinetochores/condition. CT vs CycA KD p=0.01, CT vs MYPT1 KD p=0.001; unpaired, two-tailed t-test).

DOI: https://doi.org/10.7554/eLife.29303.016

interaction of MYPT1 and Plk1 during early mitosis. This priming would be expected to diminish once Cyclin A is degraded as cells reach metaphase, which fits a need in prometaphase to titrate the k-MT stabilizing effect of Plk1 activity.

We also provide evidence that the extent of Plk1 substrate self-priming is influenced by the presence of Cyclin A or MYPT1. Self-priming and Cdk1-dependent priming of Plk1 substrates were previously considered as independent, unrelated processes (*Lee et al., 2014*; *Schmucker and Sumara, 2014*). Our data shows that MYPT1 provides a molecular link between these two Plk1 substrate priming pathways. MYPT1 is primed for Plk1 interactions by Cyclin A/Cdk1 and subsequently acts to dampen Plk1 activity at kinetochores by regulating the extent of Plk1 self-priming of PBIP1 (*Figure 6*). This finding implies that the autonomous, feed-forward mechanism of self-priming of Plk1 substrates can be constrained by the Cdk1-dependent priming of substrates. With respect to pools of Plk1 acting at kinetochores (*Lera et al., 2016*), this constraint is important to prevent excessive activation of Plk1 which could hyper-stabilize k-MT attachments and impair the efficient correction of k-MT attachment errors.

Finally, our data indicate extensive cross-regulation between the Cdk1-, Aurora B kinase-, and Plk1-dependent mitotic signaling networks in early mitosis. Cyclin A/Cdk1 promotes the destabilization of k-MT attachments in prometaphase (*Kabeche and Compton, 2013*; *Zhang et al., 2017*), and

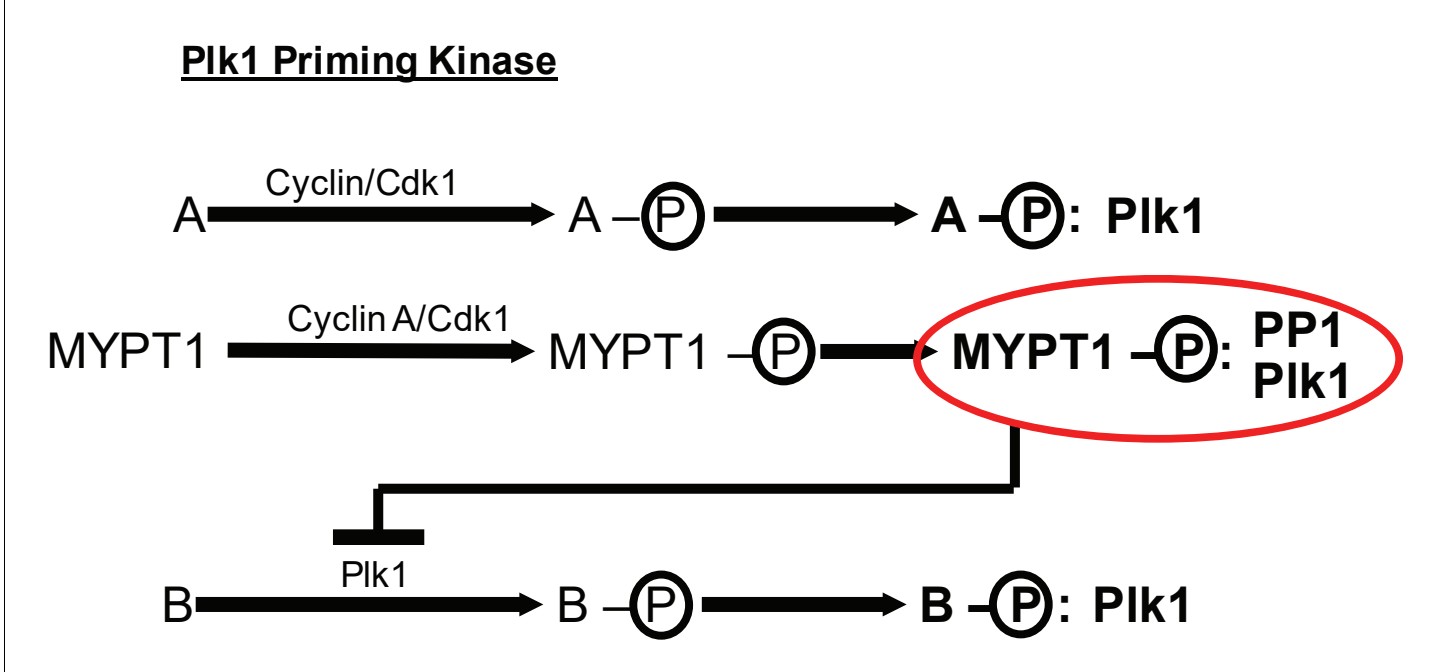

**Figure 6.** MYPT1 cross-regulates Cyclin A/Cdk1-dependent priming and self-priming of Plk1 substrates during mitosis. Cyclin/Cdk1 has an established role in catalyzing the priming phosphorylation of substrates to promote Plk1 binding as depicted for hypothetical substrate 'A'. MYPT1 is a specific substrate shown to be sensitive to Cyclin A/Cdk1 catalyzed priming phosphorylation for interaction with Plk1. In turn, MYPT1 recruits PP1 to dampen Plk1 which reduces the level of self-priming for hypothetical substrate 'B'.

DOI: https://doi.org/10.7554/eLife.29303.017

it can do so by directly phosphorylating substrates or indirectly by influencing the activities of other mitotic kinases. In the presence of Cyclin A, we observe increased phosphorylation of the activating T-loop on Aurora B kinase which plays a well characterized role in destabilizing k-MT attachments to promote error correction (*Welburn et al., 2010*; *Salimian et al., 2011*; *DeLuca et al., 2011*; *Lampson and Cheeseman, 2011*; *Cimini et al., 2006*; *Chan et al., 2012*), and decreased phosphorylation of the activating T-loop on Plk1 which plays a role in stabilizing k-MT attachments in early mitosis (*Lénárt et al., 2007*; *Liu et al., 2012*; *Moutinho-Santos et al., 2012*; *Suijkerbuijk et al., 2012*; *Li et al., 2010*; *Hood et al., 2012*; *Maia et al., 2012*). These data underscore the importance of systems-based views of mitotic signaling networks in that k-MT attachment stability at each phase of mitosis is a readout of integrated, opposing inputs from multiple kinases and phosphatases.

## Materials and methods

### Cell lines

In the present study, four mammalian cell lines were used h-TERT immortalized Retinal Pigment Epithelial 1 (RPE1) ATCC CRL-4000, RRID:CVCL_4388, Human Bone Osteosarcoma Epithelial (U2OS) ATCC HTB-96, RRID:CVCL_0042, RPE1s stably expressing photoactivatable GFP-tubulin (PA RPE1s) (*Kabeche and Compton, 2013*), and U2OS cells stably expressing photoactivatable GFP-tubulin (PA U2OS) (*Kabeche and Compton, 2013*). All cell lines used were previously characterized (RPE1 and U2OS cell lines were obtained from ATCC, authenticated with STR profiling; stable PA cell lines were generated from these cell lines as described in [*Kabeche and Compton, 2013*]), no further authentication was done for these studies; all cell lines used are validated as mycoplasma-free and were grown at 37°C in a humidified atmosphere with 5% $CO_2$.

### Cell culture

RPE1 (ATCC, CRL-4000) cells were grown in Dulbecco's modified Eagle's medium (DMEM; Invitrogen); U2OS (ATCC, HTB-96) and photoactivatable GFP-tubulin-expressing (PA)-U2OS cells were grown in McCoy's 5A (modified) Medium (McCoy's; Invitrogen) supplemented with 10% FBS (Hyclone), 250 µg/L Amphotericin B (Sigma Aldrich), 50 U/mL penicillin (Mediatech), and 50 µg/mL streptomycin (Mediatech). PA-U2OS cell media was further supplemented with 1 mg/mL G418 (InvivoGen).

For the phosphoproteomics experiment, RPE1 cells were switched from supplemented DMEM to high-glucose DMEM free of glutamine, lysine and arginine (Gibco/Corning; A1443101) containing 10% dialyzed FBS (Gibco/Corning) and 250 mg/L proline and 150 mg/L heavy or light arginine and lysine. Testing for incorporation and conversion of isotopically labeled amino acids was conducted.

### Cell transfection

Short interfering RNA (siRNA) transfections were performed using Oligofectamine (Invitrogen), and cells were collected for analysis 48 hr after transfection. RNA duplexes were purchased from ThermoFisher and used at a concentration of 200 nM. Cyclin A-*CCNA2* (Silencer Select Validated; 5'-GAUAUACCCUGGAAAGUCUtt-3'), Ambion Cat#4390825; ID: s2513. MYPT1-*PPP1R12A* (Silencer Select Validated; 5'- GCAGUACCUCAAAUCGUUUtt-3'), Ambion Cat#4390825; ID: s9237

### Mutagenesis

Full-length MYPT1 plasmid was a gift from Erika Lutter (Oklahoma State University), cloned as previously described (*Lutter et al., 2013*). Primers for mutagenesis were designed using New England Biolabs NEBaseChanger and purchased from Integrated DNA Technologies. Using the New England Biolabs Q5-Site Directed Mutagenesis Kit, the MYPT1 plasmid was mutated at the Ser472:Ser473 sequence to generate three MYPT1 mutants: MYPT1Ser472:Asp473, MYPT1Asp472:Asp473, and MYPT1Ser472:Ala473. These mutant MYPT1 plasmids were then transformed into high-efficiency NEB 5-alpha Competent E.Coli for amplification, and subsequently isolated using Qiagen QIAPrep Spin Mini-Prep and Maxi-Prep Kits. Isolated plasmids were sequenced to verify successful mutagenesis before being transfected into human cells. Photoactivatable U2OS cells were transfected with either full-length MYPT1 plasmid, MYPT1-437A plasmid, MYPT1-473D plasmid, or MYPT1-472:473DD plasmid for 23 hr. Cells were released into G418 selection media for 12–24 hr before

photoactivation. Primers used: F(TTCAGCTTCAGCTCCCAGACTTTCCTCC), R(CGTGTAACACC TGCAGTATC)

F(ACGTTCAGCTGCAGCTCCCAGAC), R(GTAACACCTGCAGTATCTTTTTCTTTCTG)
F(ACGTTCAGCTGACGATCCCAGAC), R(GTAACACCTGCAGTATCTTTTTC)
F(TTCAGCTTCAGATCCCAGACTTTCCTCC), R(CGTGTAACACCTGCAGTATC)

## Western blotting

Cells were lysed and boiled in SDS Sample Buffer (1MTris, 50% glycerol, 10% SDS, 0.5% bromophenol blue, β-mercaptoethanol) for 10 min, and then loaded on SDS-PAGE gels. Separated proteins were transferred to nitrocellulose membranes (Immobilon-P; Merck Millipore Ltd.). Membranes were incubated with primary antibody in a 2% TBS-Tween-dried milk solution either 3 hr at RT or overnight at 4°C on a rotating plate. Following a 5 min wash in 0.5% TBS-Tween, membranes were incubated for 45 min-1hr at RT on a rotating plate with horseradish peroxidase secondary in 2% TBS-Tween-dried milk solution. Immunoblots were detected using Lumiglow (KPL).

Quantification was done by measuring inverted-color average pixel intensity using fixed-sized area around the bands of interest which were then background-corrected by subtracting an average of several measured areas of identical size at nonspecific regions of the membrane. Quantifications were done using Fiji (ImageJ) software. The results were normalized to the loading control signal for each condition.

## Calcium stable assay

U2OS cells were grown on coverslips; untreated cells and cells depleted of MYPT1 via siRNA were treated with $CaCl_2$ buffer (100 mM PIPES, 1 mM $MgCl_2$, 1 mM $CaCl_2$, 0.5% Triton-X, pH = 6.8) for 5 min and subsequently fixed with 1% glutaraldehyde in PBS for 10 min. Coverslips were then treated with $NaBH_4$ for 10 min x2, and then stained using the regular immunofluorescence protocol as described below.

## Chromosome spreads

siRNA depletion of Cyclin A was accomplished via transfection as described above using U2OS cells. 48 hr after transfection, U2OS CT and KD cells were arrested overnight in media containing 3.33 μM Nocodazole. Mitotic shake-offs were performed and mitotic cells were incubated for 10 min in hypotonic RSB solution [10 mM Tris-HCl pH 7.5, 10 nM NaCl, 3 mM $MgCl_2$ in $ddH_2O$] as previously described, before being spun at 1600 RPM onto poly-D-lysine coated coverslips. 3.7–3.9% PFA for 10 min was used for fixation, after which IF staining was performed as described below.

## Immunofluorescence and image processing

For visualization of most antibodies, cells were fixed with 3.5% paraformaldehyde (PFA) for 15 min, washed with Tris-buffered saline with 5% bovine serum albumin (TBS-BSA) containing 0.5% Triton-X-100 for 5 min, and then rinsed in TBS-BSA for 5 min. Primary antibodies were diluted in stock solutions of TBS-BSA containing 0.1% Triton-X-100. Coverslips were incubated with primary Ab for 1–3 hr at room temperature, washed for 2 × 10 min in 0.1%Triton-TBS-BSA, and then incubated with secondary antibody and DAPI diluted in 0.1% Triton-TBS-BSA. Two subsequent 10 min washes were done in 0.1% Triton-TBS-BSA followed by a 10 min wash in TBS-BSA. Coverslips were mounted on slides using ProLong Gold antifade reagent (Molecular Probes). For phosphoantibodies, pre-extraction was done prior to PFA fixation using MTSB and 1% Triton-X-100-TBS-BSA as follows: 1 min MTSB, 2 min MTSB +1% Triton, 2 min MTSB.

Images were acquired with Orca-ER Hamamatsu cooled charge-coupled device camera mounted on an Eclipse TE 2000-E Nikon microscope. 0.2 or 0.3 μm optical sections in the z-axis were collected with a plan Apo 60 × or 100 × 1.4 numerical aperture oil immersion objective at room temperature.

**Antibodies**

| | | |
|---|---|---|
| Mouse monoclonal anti-Actin (WB at 1:4000) | Seven hills bioreagents | Cat #LMAB-C4 |

*Continued on next page*

*Continued*

**Antibodies**

| | | |
|---|---|---|
| Human anti-ACA (anti-centromere antibody; IF at 1:1000) | Gift, Geisel School of Medicine | n/a |
| Rabbit polyclonal anti-Aurora B (IF/WB 1:1000) | Novus Biologicals | Cat# NB100-294 |
| Mouse monoclonal anti-Cyclin B (WB 1:1000) | Santa Cruz | Cat# SC-245 |
| Rabbit polyclonal anti-Cyclin A (WB 1:1000) | Santa Cruz | Cat# SC-751 |
| Rabbit polyclonal anti-MYPT1 (IF/WB 1:1000); | Santa Cruz | Cat# SC-25618 |
| Mouse anti-MYPT1 pSer473 (IF/WB 1:1000) | Gift, Fumio Matsumura; *Yamashiro et al. (2008)* | n/a |
| Rabbit polyclonal anti-Nek2 (WB 1:1000); | Santa Cruz | Cat# SC-33167 |
| Rabbit monoclonal anti-Chk2 pT68 (WB 1:1000); | Cell Signaling | Cat# C13C1 |
| Rabbit polyclonal anti-MLF1 Interacting Proten/PBIP1 pThr78 (IF/WB 1:500); | Abcam | Cat# Ab34911 |
| Rabbit polyclonal anti-Plk1 (IF/WB 1:1000); | Bethyl Laboratories | Cat# A300-251A |
| Mouse monoclonal anti-Plk1 pThr210 (IF/WB 1:500); | Abcam | Cat# Ab39068 |
| Mouse monoclonal anti-Securin (WB 1:1000) | Abcam | Cat# AB3305 |
| Mouse monoclonal anti- α-Tubulin (DM1α) (IF at 1:3000, WB at 1:4000). | Sigma Aldrich | Cat# T6199 |
| Mouse monoclonal anti-Hec1 (IF at 1:500) | Novus | Cat# NB100-338 |
| Secondary antibodies used for IF were goat anti-mouse; goat anti-human; goat anti-rabbit; donkey anti-rabbit; donkey anti-mouse; 488, 568, 647 (IF at 1:1000) | Alexa Fluor Life Technologies | Cat# A21445; A21202; A11008; A11013; A31573; A11014; A11032 |
| Secondary antibodies used for WB were goat anti-rabbit IgG (H + L)-HRP conjugate and goat anti-mouse IgG (H + L)-HRP conjugate (WB at 1:1000). | Bio-Rad | Cat# 170–6515, 170–6516 |

## Monastrol washout and mitotic error rates

U2OS cells were transfected with siRNA to knock down MYPT1 as described above. At 48 hr after transfection, CT and KD cells were incubated in media containing 100 μM Monastrol for 4 hr, and then released for 1–2 hr (until mitotic wave) in normal media. Cells were fixed in PFA and IF staining was performed as described above. N ≥ 300 anaphases per condition from two independent experiments were analyzed for mitotic errors.

## Inter-kinetochore distance and spindle length

Inter-kinetochore distance was measured in images of fixed cells by distance between Hec1 staining in sister kinetochores in untreated (CT) cells and cells depleted of MYPT1 via si-RNA (MYPT1 KDs). n ≥ 100 kinetochore pairs per condition. Spindle length was measured in images of fixed cells by distance from the centroid of maximal tubulin staining intensity from centrosome to centrosome. Quantifications were done using Fiji (ImageJ) software.

## Photoactivation

k-MT attachment stability was assessed by measuring microtubule turnover rates in live U2OS cells stably expressing GFP-tagged α-tubulin (plasmid provided by Alexey Khodjakov). Differential interference contrast (DIC) microscopy was used to identify prometaphase and metaphase mitotic cells. Fluorescent images were acquired using Quorum WaveFX-X1 spinning disc confocal system (Quorum Technologies) equipped with Mosaic digital mirror for photoactivation (Andor Technology) and Hamamatsu ImageEM camera. Prometaphase or metaphase determination was made based on chromosome alignment as visualized by DIC. Microtubules across one half-spindle were laser-illuminated over a defined region, and fluorescence images were subsequently captured every 15 s for a time-course of 4 min with a 100 × oil immersion 1.4 numerical aperture objective.

## Stable isotope labeling

RPE1 cells were switched from supplemented DMEM to the growth medium for SILAC experiments as detailed above, which was based upon modified media as previously described (*Ong et al., 2002*; *Kettenbach et al., 2011*), and was sterile-filtered before use. Testing for incorporation was conducted as previously described (*Kettenbach et al., 2011*)

## Generating differential cyclin A conditions

One population of RPE1s was grown exclusively in heavy-labeled isotope media, while the other was grown exclusively in light-labeled isotope media. After incorporation, both heavy and light-labeled RPE1s were synchronized in mitosis using a double-thymidine block (2 mM thymidine for 16 hr, 9 hr release into media, 2 mM thymidine for 15 hr, 7 hr release into media). 7 hr after second release, cells were arrested into 100 nM taxol-media for 2 hr. Mitotic shake-off was performed on both heavy-labeled and light-labeled cells. Heavy-labeled cells were pelleted and flash-frozen. Light-labeled cells were collected into 50 mL conical tubes and incubated in 100 nM taxol-media for an additional 10 hr before being pelleted and flash-frozen.

## Phosphopeptide enrichment

Phosphopeptide purification was performed as previously described (*Kettenbach and Gerber, 2011*). Briefly, peptides were resuspended in 2M lactic acid in 50% ACN 'binding solution.' Titanium dioxide microspheres were added to select for phosphopeptides. Microspheres were washed twice with binding solution and three times with 50% ACN/0.1% TFA. Peptides were eluted with 50 mM $KH_2PO_4$ (adjusted to pH 10 with ammonium hydroxide). Peptide elutions were combined, quenched with 50% ACN/5% formic acid, dried, and desalted on a μHLB OASIS $C_{18}$ desalting plate (Waters).

## SCX chromatography

Phosphopeptides were resuspended in SCX buffer and separated per injection on an SCX column as previously described. (*Kettenbach and Gerber, 2011*). Fractions were collected, dried, and desalted on a μHLB OASIS $C_{18}$ desalting plate (Waters).

## Quantification and statistical analysis

### Proteomics analysis

LC-MS/MS analyses for peptides and phosphopeptides were performed using a Q-Exactive Plus mass spectrometer (Thermo Fisher Scientific) equipped with an Easy-nLC1000 (Thermo Fisher Scientific) and nanospray source (Thermo Fisher Scientific). Peptides and phosphopeptides were redissolved in 5% ACN/1% formic acid and analyzed as previously described (*Petrone et al., 2016*). Raw data were searched using COMET (release version 2014.01, RRID:SCR_011925) in high resolution mode (*Eng et al., 1994*) against a target-decoy (reversed) (*Elias and Gygi, 2007*) version of the human proteome sequence database (UniProt; RRID:SCR_002380 downloaded 2/2013, 40482 entries of forward and reverse protein sequences) with a precursor mass tolerance of ±1 Da and a fragment ion mass tolerance of 0.02 Da, and requiring fully tryptic peptides (K,R; not preceding P) with up to three mis-cleavages. Static modifications included carbamidomethylcysteine and variable modifications included: oxidized methionine, heavy lysine and arginine, phosphorylated serine, threonine, and tyrosine. Searches were filtered using orthogonal measures including mass measurement accuracy (±3 ppm), Xcorr for charges from +2 through +4, and dCn targeting a <1% FDR at the peptide level (Wang et al., 2012). The probability of phosphorylation site localization was assessed using PhosphoRS (*Taus et al., 2011*). Quantification of LC-MS/MS spectra was performed using MassChroQ (*Valot et al., 2011*). Phosphopeptide ratios were adjusted for mixing errors based on the median of the $\log_2$ H/L distribution and corrected for differences in protein abundance.

Significance of $\log_2$ phosphopeptide fold-change was determined by two-tailed Student's t test assuming unequal variance. Protein-protein interactions of proteins belonging to phosphopeptides were determined using the STRING database and analyzed in Cytoscape RRID:SCR_003032 (*Shannon et al., 2003*) (*Saito et al., 2012*). Edges present protein-protein interactions based on the STRING database. Densely connected clusters were identified using STRING and ClusterONE (*Bader and Hogue, 2003*) in Cytoscape.

Complete phosphoproteome dataset can be found in *Supplementary file 1*. Datasets filtered by kinase consensus motifs can be found in *Supplementary files 1*, *2,* and *3*.

The mass spectrometry proteomics data have been deposited to the ProteomeXchange Consortium via the PRIDE partner repository with the dataset identifier PXD008316.

### Photoactivation

Fluorescence dissipation after photoactivation (FDAPA) was calculated as previously described. Briefly, using Quorum WaveFX software, mean pixel intensities were quantified within a rectangular area surrounding the region of highest fluorescence intensity and background substracted using an equally-sized area from the non-activated half-spindle at the same position. Quantified intensities were background-corrected and corrected for photobleaching and then normalized to the first time-point after photoactivation for each individual cell. Microtubule turnover rates were then calculated using MatLab (MathWorks) software RRID:SCR_001622 to fit the corrected fluorescence intensities to a two-variable exponential curve [$A1 \times \exp(k_1 t) + A2 \times (\exp(k_2 t))$], where t = time, A1 = non k-MT population of microtubules with decay rate $k_1$, A2 = k MT population of microtubules, with decay rate $k_2$ corresponding. Solving the equation for the two-variable exponential curve yields the rate constants and fraction of microtubules pertaining to each of the populations (non-k-MT/fast turnover population and k-MT/slow turnover population). The turnover half-life was then calculated as $\ln_2/k$ for each population of microtubules across each of the experimental conditions.

### Statistical analysis

For anaphase error rates, $\geq 800$ anaphases per condition from two independent experiments were quantified. Chi square analysis was performed. For photoactivation, $\geq 15$ mitotic cells per condition from $\geq 3$ independent experiments were quantified. $R^2$ values for photoactivation were generated using MatLab software. For quantification of phospho-PBIP1 T78, phospho-Plk1 T210, and Plk1 levels at kinetochores, $\geq 100$ kinetochores per condition from two experiments were quantified. Statistical analysis was performed applying Student's two-tailed t-test (unpaired) using GraphPad Prism software RRID:SCR_002798.

### Contact for reagent and resource sharing

Further information and requests for resources and reagents should be directed to and will be fulfilled by the Corresponding Author, Duane Compton (Duane.Compton@dartmouth.edu).

## Acknowledgements

We would like to thank all members of the Compton lab for helpful discussions, critical feedback, and support. We thank Fumio Matsumura for kindly providing the MYPT1 pSer473 antibody. We also thank Erika Lutter for kindly providing the MYPT1 plasmid. This work was supported by National Institutes of Health grants GM051542 (DAC), and GM119455 (ANK).

## Additional information

### Funding

| Funder | Grant reference number | Author |
| --- | --- | --- |
| National Institute of General Medical Sciences | GM119455 | Arminja N Kettenbach |
| National Institute of General Medical Sciences | GM051542 | Duane A Compton |

The funding agency had no role in study design, data collection and interpretation, or the decision to submit the work for publication.

### Author contributions

Ana Maria G Dumitru, Conceptualization, Validation, Investigation, Methodology, Writing—original draft, Project administration, Writing—review and editing; Scott F Rusin, Formal analysis, Validation,

Investigation, Methodology; Amber E M Clark, Investigation; Arminja N Kettenbach, Conceptualization, Data curation, Supervision, Funding acquisition, Project administration, Writing—review and editing; Duane A Compton, Conceptualization, Supervision, Writing—original draft, Project administration, Writing—review and editing

### Author ORCIDs

Ana Maria G Dumitru (iD) http://orcid.org/0000-0002-8114-051X
Arminja N Kettenbach (iD) http://orcid.org/0000-0003-3979-4576
Duane A Compton (iD) http://orcid.org/0000-0002-4445-9118

### Decision letter and Author response

Decision letter https://doi.org/10.7554/eLife.29303.025
Author response https://doi.org/10.7554/eLife.29303.026

## Additional files

### Supplementary files

• Supplementary file 1. Mitotic phosphoproteome dataset unfiltered and filtered by Cdk1 consensus motif Complete phosphoproteome dataset, along with dataset filtered by Cdk1 minimum consensus motif. Raw data were searched using COMET in high resolution mode against a target-decoy (reversed) version of the human proteome sequence database, filtered, and quantified, and phosphopeptide ratios were adjusted as described in the Methods section of the text. Significance of $\log_2$ phosphopeptide fold-change was determined by two-tailed Student's t test assuming unequal variance. Dataset was also filtered by the Cdk1 minimum consensus motif (pS/pT-P; (*Nigg, 1993*; *Moreno and Nurse, 1990*; *Songyang et al., 1994*)), listed in the second tab of the file.
DOI: https://doi.org/10.7554/eLife.29303.018

• Supplementary file 2. Phosphoproteome dataset filtered by Aurora B consensus motif Dataset filtered by the Aurora B kinase consensus motif (R/K – X – pS/pT) (*Kim et al., 2010*; *Meraldi et al., 2004*).
DOI: https://doi.org/10.7554/eLife.29303.019

• Supplementary file 3. Phosphoproteome dataset filtered by Plk1 consensus motif Dataset filtered by the Plk1 kinase consensus motif (D/E-X-pS/pT) (*Nakajima et al., 2003*; *Grosstessner-Hain et al., 2011*).
DOI: https://doi.org/10.7554/eLife.29303.020

• Transparent reporting form
DOI: https://doi.org/10.7554/eLife.29303.021

### Major datasets

The following dataset was generated:

| Author(s) | Year | Dataset title | Dataset URL | Database, license, and accessibility information |
|---|---|---|---|---|
| Ana Maria G Dumitru, Scott F Rusin, Arminja N Kettenbach, Duane A Compton | 2017 | Cyclin A/Cdk1 modulates Plk1 activity in prometaphase to regulate k-MT attachment stability | https://www.ebi.ac.uk/pride/archive/projects/PXD008316 | Publicly available at the PRIDE database (accession no. PXD008316) |

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
