## [Decision Letter]

Thank you for submitting your article "Cyclin A/Cdk1 modulates Plk1 activity in prometaphase to regulate kinetochore-microtubule attachment stability" for consideration by *eLife*. Your article has been reviewed by two peer reviewers, and the evaluation has been overseen by a Reviewing Editor and Andrea Musacchio as the Senior Editor.

The reviewers have discussed the reviews with one another and the Reviewing Editor has drafted this decision to help you prepare a revised submission. All three of them were troubled by several issues that we would insist be addressed to more firmly allow the primary conclusions concerning cyclin A regulation of the MYPT1 phosphatase as a means to control the Plk1 kinase at kinetochores and its influence on kinetochore microtubule stabilization. You may be able to easily deal with some of the points where they are confused. Others will require further experimentation – and they make specific suggestions below. We would be glad to consider a revision if you are able to add these.

Summary:

Now to the specifics. Your manuscript identifies cyclin A targets in prometaphase and then demonstrates that the phosphatase targeting subunit MYPT1 regulates Plk1 function to alter kinetochore-microtubule attachments in mitosis. This adds to your previous work describing a global switch in kinetochore-microtubule stability between prometaphase and metaphase. That earlier work attributed this switch to the destruction of Cyclin A, and the subsequent inactivation of CyclinA/CDK1 complexes, but did not identify the key Cyclin A/CDK substrates. You do this here using quantitative MS approaches to identify specific cyclin A targets during prometaphase. Many cyclin A phospho-sites are identified that cluster into groups regulating key mitotic kinases, such as Plk1 and Aurora B. Because MYPT1 has a known function regulating Plk1, its role in kinetochore-microtubule attachments is specifically tested. Reduction in MYPT1 or cyclin A is found to increase Plk1 at kinetochores and this suggest an altered interaction between MYPT1 and Plk1. identifying mitotic proteins/peptides whose phosphorylation is increased in the presence of high levels of Cyclin A. One of these proteins is Mypt1, a PP1 regulatory subunit that negatively regulates Plk1. You confirm that Mypt1 phosphorylation leads to decreased PLK1 activity, and further demonstrate that depletion of Mypt1 leads to defects in kinetochore-microtubule turnover in both prometaphase and metaphase. This leads to a major conclusion that Cyclin A/CDK1 regulation of Mypt1, through Mypt1's regulation of PLK1, is largely responsible for the switch in K-MT stability from prometaphase to metaphase.

We agree that identifying the key targets of CyclinA/Cdk1 is of interest. Our feeling, however, is that the conclusions are generally overstated as they rely on the use of knockdowns of cyclin A and MYPT1 and many targets are affected when these genes are knocked down, limiting the conclusiveness of any potentially direct interactions.in regard to identifying the phosphorylation event responsible. We are not convinced that Cyclin A/CDK1 phosphorylation of MYPT1 is the causative event.

Essential revisions:

1) To make the conclusion that the MYPT1 phosphorylation at S473 regulates kinetochore-microtubule attachments via PLk1 and that it is required to maintain Plk1 kinetochore levels, experiments on the cyclin A phosphorylation site mutant in MYPT1 are essential.

2) You report that depletion of MYPT1 leads to increased kinetochore-MT stability in prometaphase (compared to normal PM values) and decreased kinetochore-MT stability in metaphase – all the way down to control PM values. We have a couple of questions:

-Why are the kMTs unstable in metaphase after Mypt1 depletion? You argue that increased PLK1 activity leads to prematurely stabilized kMTs, but what is the explanation for significantly decreased kMT stability in metaphase?

– If these changes are due to decreased MYPT1 phosphorylation, and subsequent increased PLK1 activity, then why isn't this same trend (namely altered turnover in metaphase) also observed in response to Cyclin A inhibition (Kabeche 2013)?

– Given the very high turnover of metaphase kinetochore-MTs after MYPT1 knockdown (down to PM levels), how are chromosomes properly aligning? Are there delays in alignment? Even more puzzling -- how are these cells silencing the checkpoint in the presence of globally unstable kMT attachments?

3) The phenotypic outcome resulting from inhibition of Cyclin A in the Kabeche paper (2013) was an increase in chromosome segregation errors due to hyper-stabilization of k-MTs (in otherwise unperturbed cells). If the major target of CyclinA/CDK1 for regulation of k-MTs is MYPT1 (which, when phosphorylated, decreases PLK1 activity to destabilize k-MTs), the expectation is that depletion of MYPT1 should mimic the phenotype observed after Cyclin A depletion. However, this is not the case.

4) A major conclusion is that priming phosphorylation of Mypt1 by CycA/CDK promotes Plk1 binding in early mitosis to promote high kMT turnover. You state, "Cyclin A/Cdk1-catalyzed priming of MYPT1 provides an inherent bias favoring the interaction of MYPT1 and Plk1 to early mitosis. This priming would be expected to diminish once Cyclin A is degraded as cells reach metaphase, which fits a need in prometaphase to titrate the k-MT stabilizing effect of Plk1 activity." A test is needed to see if this is the case using the MYPT1 phospho-specific antibody to determine when MYPT1 is normally phosphorylated during mitosis. Are the temporal phosphorylation dynamics consistent with their conclusion?

5) In regard to the turnover data, you use U2OS cells. The Compton group has demonstrated that such CIN cell lines exhibit significantly deviant kMT turnover/dynamics, thus it doesn't seem to make sense why this particular cell line has been chosen for the analyses. Are similar results found in RPE1 cells, which have been used extensively in the Compton lab for such analyses?

6) A new finding here is that CyclinA/CDK (vs. CycB/CDK or a combination of CycA/CDK CycB/CDK) phosphorylates Mypt1 to promote Plk1 binding and inactivation. The rationale for this conclusion is that in cells depleted of Cyclin A, Cyclin B levels do not change but Mypt1 phosphorylation is decreased. In Figure 2, it looks by eye as though Cyclin B levels are decreased after Cyclin A knockdown. Quantification here would help substantiate the claim.

7) In Cyclin A "high" cells, the prediction would be that PLK1 phosphorylation at Thr210 would decrease (high Cyclin A → high Mypt1 phosphorylation → decreased PLK1 phosphorylation). Is this the case? From the statement in subsection “Intersection of mitotic signaling networks”, it seems to be the contrary: you report that most PLK1 sites either did not exhibit altered phosphorylation or exhibited increased phosphorylation.

8) You demonstrate that turnover of K-MTs is altered in MYPT1 depleted cells. Do these changes in turnover reflect defects in kinetochore-MT polymerization/depolymerization dynamics (e.g. are kinetochore oscillatory movements perturbed?) or defects in kinetochore-MT attachment strength (e.g. are pulling forces changed? Do cells exhibit altered cold-resistant kMTs?). Is this consistent with previously described roles for Plk1 in k-MT stabilization?

---

## [Author Response]

Essential revisions:1) To make the conclusion that the MYPT1 phosphorylation at S473 regulates kinetochore-microtubule attachments via PLk1 and that it is required to maintain Plk1 kinetochore levels, experiments on the cyclin A phosphorylation site mutant in MYPT1 are essential.

We thank the reviewers for raising this specific point. We have generated three different mutant versions of MYPT1: MYPT1473Ser→Ala, MYPT1473Ser→Asp, and MYPT1472:473-SerSer→AspAsp. We show that the alanine mutant prematurely stabilizes k-MT attachments in prometaphase and destabilizes k-MT attachments in metaphase. This resembles the effects observed in MYPT1-depleted cells indicating that the alanine mutant acts in a dominant negative manner to disrupt MYPT1 function and its role in regulating k-MT attachment stability. We created both the single (473) and double (472 & 473) aspartic acid mutants to account for the possibility of phosphorylation at one or both of the adjacent serine residues. In contrast to the alanine mutant, both aspartic acid mutants prevent the stabilization of k-MT attachments that ordinarily occurs at metaphase. This is the expected outcome for an irreversible phosphor-mimic at this residue that continues to exert a destabilizing effect on k-MT attachments into metaphase. The new data generated using these mutant forms of MYPT1 is shown in Figure 4 and Figure 4—figure supplement 4.

Overall, we believe that these results support our conclusion that the Cyclin A/Cdk1-mediated phosphorylation of MYPT1 Serine 473 is directly involved in regulating the attachment stability of k-MT attachments during mitosis. We believe that these new data strengthen our conclusion and thank the reviewers for this suggestion.

2) You report that depletion of MYPT1 leads to increased kinetochore-MT stability in prometaphase (compared to normal PM values) and decreased kinetochore-MT stability in metaphase – all the way down to control PM values. We have a couple of questions:-Why are the kMTs unstable in metaphase after Mypt1 depletion? You argue that increased PLK1 activity leads to prematurely stabilized kMTs, but what is the explanation for significantly decreased kMT stability in metaphase?

We thank the reviewers for their thoughtful question. We agree that this is an intriguing result, but we feel that it is difficult to provide a satisfactory answer given the current understanding for how all the various molecules work together to generate the desired stability of k-MT attachments in different phases of mitosis. It is possible that the biochemical ‘state’ of metaphase cells lacking MYPT1 reflects on the overall stability of k-MT attachments differently than in prometaphase. As such, we suggest that the metaphase destabilizing effect is likely the result of skewing the balance between Plk1-mediated phosphorylation of k-MT stabilizers and destabilizers, possibly secondary to the localization effects of Cyclin A phosphorylation on MYPT1 kinetochore recruitment, as we show in Figure 2. For emphasis, we cite the evidence in the literature for somewhat paradoxical Plk1-mediated activation of both k-MT stabilizers and destabilizers at kinetochores (listed below). Ultimately, understanding this issue will take substantial additional experimentation that would go well beyond the scope of this manuscript which already includes substantial phosphoproteomic screening data and validation of that screen through the targeted analysis of a specific substrate.

Lenart P, et al. The small-molecule inhibitor BI2536 reveals novel insights into mitotic roles of polo-like kinase 1. Curr. Biol. 2007;17:304–315.

Elowe S, Hummer S, Uldschmid A, Li X, Nigg EA. Tension-sensitive Plk1 phosphorylation on BubR1 regulates the stability of kinetochore microtubule interactions. Genes Dev. 2007;21:2205–2219.

Liu D, Davydenko O, Lampson MA. Polo-like kinase 1 regulates kinetocohore-microtubule dynamics and spindle checkpoint silencing. J. Cell Biol. 2012;198:491–499.

Lampson MA, Kapoor M. The human mitotic checkpoint protein BubR1 regulates chromosome-spindle attachments. Nat. Cell Biol. 2005;7:93–98.

Manning AL, et al. The kinesin-13 proteins Kif2a, Kif2b, and Kif2c/MCAK have distinct roles during mitosis in human cells. Mol. Biol. Cell. 2007;18:2970–2979.

Hood EA, Kettenbach AN, Gerber SA, Compton DA. Plk1 regulates the kinesin-13 protein Kif2b to promote faithful chromosome segregation. Mol. Biol. Cell. 2012;23:2264–2274.

Lera, R., Potts, G., Suzuki, A., Johnson, J., Salmon, E., Coon, J., & Burkard, M. (2016).Decoding Polo-like kinase 1 signaling along the kinetochore-centromere axis. Nature Chemical Biology, 411-418.)

– If these changes are due to decreased MYPT1 phosphorylation, and subsequent increased PLK1 activity, then why isn't this same trend (namely altered turnover in metaphase) also observed in response to Cyclin A inhibition (Kabeche 2013)?

Our data shows that Cyclin A/Cdk1 phosphorylates more substrates than just MYPT1 (Figure 1 and Figure 1—figure supplement 1 and Supplementary file 1–Supplementary file 3). Indeed, we identified over 100 phosphopeptides which were sensitive to differential Cyclin A expression in our experimental conditions. As such, it stands to reason that the observed effects reported in the Kabeche 2013 are the global effect of Cyclin A inhibition, rather than the isolated effect of one branch of regulation of k-MT stability, which we think we provide strong evidence for in this paper.

– Given the very high turnover of metaphase kinetochore-MTs after MYPT1 knockdown (down to PM levels), how are chromosomes properly aligning? Are there delays in alignment? Even more puzzling -- how are these cells silencing the checkpoint in the presence of globally unstable kMT attachments?

We thank the reviewers for their keen observations and insightful questions. First, chromosome alignment defects and delays have been previously reported in Hela cells depleted of MYPT1.

Yamashiro, S., Yamakita, Y., Totsukawa, G., Goto, H., Kaibuchi, K., Ito, M.,... Matsumura, F. (2008). Myosin Phosphatase-Targeting Subunit 1 Regulates Mitosis by Antagonizing Polo-like Kinase 1. Developmental Cell, 787-797.

Totsukawa, T., Yamakita, Y., Yamashiro, S., Hosoya, H., Hartshorne, D., & Matsumura, F. (1999). Activation of myosin phosphatase targeting subunit by mitosis-specific phosphorylation. J Cell Biology, 735-744.

Second, we noted chromosome alignment defects in RPE1 cells when we attempted to perform photoactivation experiments in MYPT1-depleted cells.

Third, our assay is designed to quantitatively measure the detachment rate of microtubules from kinetochores. The values for k-MT attachment stability reported here are well within a range that would be sufficient to support both chromosome alignment (in U2OS cells) and eventual satisfaction of the SAC. We trust the reviewers not to confuse ‘reduced attachment stability’ with ‘loss of attachments’ – those are quite different. We draw the reviewer’s attention to our measurements in U251 cells (Bakhoum et al. Curr Biol 2009) that demonstrated reversed k-MT stability (more stable prometaphase and less stable metaphase k-MT attachments). It is possible to have reduced k-MT attachment stability, yet to be able to congress chromosomes and satisfy the checkpoint and proceed into anaphase. In a separate line of inquiry, we are conducting experiments to determine the threshold for k-MT attachment stability, below which, the cells fail to satisfy the checkpoint. That line of inquiry is well beyond the scope of this current work.

3) The phenotypic outcome resulting from inhibition of Cyclin A in the Kabeche paper (2013) was an increase in chromosome segregation errors due to hyper-stabilization of k-MTs (in otherwise unperturbed cells). If the major target of CyclinA/CDK1 for regulation of k-MTs is MYPT1 (which, when phosphorylated, decreases PLK1 activity to destabilize k-MTs), the expectation is that depletion of MYPT1 should mimic the phenotype observed after Cyclin A depletion. However, this is not the case.

This is the same as comment #2, second bullet. Because Cyclin A/Cdk1 is a protein kinase with multiple mitotic substrates, it stands to reason that the loss of Cyclin A/Cdk1 would have different effects on mitotic cells than the loss of only one substrate.

4) A major conclusion is that priming phosphorylation of Mypt1 by CycA/CDK promotes Plk1 binding in early mitosis to promote high kMT turnover. You state, "Cyclin A/Cdk1-catalyzed priming of MYPT1 provides an inherent bias favoring the interaction of MYPT1 and Plk1 to early mitosis. This priming would be expected to diminish once Cyclin A is degraded as cells reach metaphase, which fits a need in prometaphase to titrate the k-MT stabilizing effect of Plk1 activity." A test is needed to see if this is the case using the MYPT1 phospho-specific antibody to determine when MYPT1 is normally phosphorylated during mitosis. Are the temporal phosphorylation dynamics consistent with their conclusion?

This is a good suggestion. Unfortunately, we were only provided with a few microliters of the pMYPT1 Ab which we exhausted with the currently described experiments. Thus, we are unable to perform the requested experiment.

5) In regard to the turnover data, you use U2OS cells. The Compton group has demonstrated that such CIN cell lines exhibit significantly deviant kMT turnover/dynamics, thus it doesn't seem to make sense why this particular cell line has been chosen for the analyses. Are similar results found in RPE1 cells, which have been used extensively in the Compton lab for such analyses?

We thank the reviewers for this suggestion. We have added new data generated using RPE1 cells (Figure 3—figure supplement 2). As the reviewers might be aware, RPE1 cells can often display somewhat different responses relative to U2OS (or other cancer cell lines), and we observed enhanced chromosome congression defects in the RPE1 cells depleted of MYPT1. We provide data showing that k-MT attachments are hyperstabilized in prometaphase, but we had difficulty assessing sufficient numbers of metaphase cells to make clear conclusions.

6) A new finding here is that CyclinA/CDK (vs. CycB/CDK or a combination of CycA/CDK CycB/CDK) phosphorylates Mypt1 to promote Plk1 binding and inactivation. The rationale for this conclusion is that in cells depleted of Cyclin A, Cyclin B levels do not change but Mypt1 phosphorylation is decreased. In Figure 2, it looks by eye as though Cyclin B levels are decreased after Cyclin A knockdown. Quantification here would help substantiate the claim.

We have revised the figures to show the relative quantities of each protein on the blots compared to the loading controls. We feel the data supports the conclusion that Cyclin A/Cdk1 is the primary kinase responsible for phosphorylating Ser473 on MYPT1. Nevertheless, we attempted to be clear in the Discussion section of the text that there may be substrates that are preferentially phosphorylated by one or the other Cyclin-associated Cdk1s or that some substrates may be equivalently phosphorylated by both.

7) In Cyclin A "high" cells, the prediction would be that PLK1 phosphorylation at Thr210 would decrease (high Cyclin A → high Mypt1 phosphorylation → decreased PLK1 phosphorylation). Is this the case? From the statement in subsection “Intersection of mitotic signaling networks”, it seems to be the contrary: you report that most PLK1 sites either did not exhibit altered phosphorylation or exhibited increased phosphorylation.

We thank the reviewers for this thoughtful question. By convention due to the way in which the isotopes correlate to the two conditions, the mass spectrometry more faithfully reports substrates that were more phosphorylated in the “high kinase activity” condition. Technical artifacts will impair conclusions drawn if one attempts to define substrates that were more phosphorylated in the low kinase activity condition without switching the isotope labeling strategy. Time and cost constraints precluded us from repeating all the necessary replicate experiments with a switched isotope labeling scheme.

8) You demonstrate that turnover of K-MTs is altered in MYPT1 depleted cells. Do these changes in turnover reflect defects in kinetochore-MT polymerization/depolymerization dynamics (e.g. are kinetochore oscillatory movements perturbed?) or defects in kinetochore-MT attachment strength (e.g. are pulling forces changed? Do cells exhibit altered cold-resistant kMTs?). Is this consistent with previously described roles for Plk1 in k-MT stabilization?

We have included additional data showing calcium-stable MT densities, inter-kinetochore distances, and spindle lengths. All of these assays are consistent with our reported data showing changes in k-MT attachment stability upon depletion of MYPT1 (Figure 4—figure supplement 2 and Figure 4—figure supplement 3). Again, these assays are measuring different features of mitotic spindles and we have purposefully focused our attention on the rate of MT detachment from kinetochores (measured using the photoactivation assay) because of the importance of MT detachment to the correction of k-MT attachment errors.